# Evaluation of an Access-Risk-Knowledge (ARK) Platform for Governance of Risk and Change in Complex Socio-Technical Systems

**DOI:** 10.3390/ijerph182312572

**Published:** 2021-11-29

**Authors:** Nick McDonald, Lucy McKenna, Rebecca Vining, Brian Doyle, Junli Liang, Marie E. Ward, Pernilla Ulfvengren, Una Geary, John Guilfoyle, Arwa Shuhaiber, Julio Hernandez, Mary Fogarty, Una Healy, Christopher Tallon, Rob Brennan

**Affiliations:** 1Centre for Innovative Human Systems, School of Psychology, Trinity College, The University of Dublin, D02 PN40 Dublin, Ireland; nick.mcdonald@tcd.ie (N.M.); bdoyle1@tcd.ie (B.D.); marie.ward@tcd.ie (M.E.W.); 2ADAPT Centre, School of Computing, Dublin City University, D09 PX21 Dublin, Ireland; lucy.mckenna@adaptcentre.ie (L.M.); junli.liang@adaptcentre.ie (J.L.); julio.hernandeztorres@dcu.ie (J.H.); rob.brennan@dcu.ie (R.B.); 3Health and Safety Unit, Dublin Fire Brigade, D02 RY99 Dublin, Ireland; john.guilfoyle@dublincity.ie (J.G.); christopher.tallon@dublincity.ie (C.T.); 4Quality and Safety Improvement Directorate, St. James’s Hospital Dublin, D08 NHY1 Dublin, Ireland; ugeary@stjames.ie (U.G.); mfogarty@stjames.ie (M.F.); uhealy@stjames.ie (U.H.); 5KTH Royal Institute of Technology, Industrial Economics and Management, 100 44 Stockholm, Sweden; pernilla.ulfvengren@indek.kth.se; 6Beacon Renal, Sandyford Business Park, D18 TH56 Dublin, Ireland; arwa@beaconrenal.ie

**Keywords:** Access Risk Knowledge (ARK), knowledge engineering platform, socio-technical systems, systems engineering, system change, risk in change, mindful governance, infection prevention control, COVID-19

## Abstract

Three key challenges to a whole-system approach to process improvement in health systems are the complexity of socio-technical activity, the capacity to change purposefully, and the consequent capacity to proactively manage and govern the system. The literature on healthcare improvement demonstrates the persistence of these problems. In this project, the Access-Risk-Knowledge (ARK) Platform, which supports the implementation of improvement projects, was deployed across three healthcare organisations to address risk management for the prevention and control of healthcare-associated infections (HCAIs). In each organisation, quality and safety experts initiated an ARK project and participated in a follow-up survey and focus group. The platform was then evaluated against a set of fifteen needs related to complex system transformation. While the results highlighted concerns about the platform’s usability, feedback was generally positive regarding its effectiveness and potential value in supporting HCAI risk management. The ARK Platform addresses the majority of identified needs for system transformation; other needs were validated in the trial or are undergoing development. This trial provided a starting point for a knowledge-based solution to enhance organisational governance and develop shared knowledge through a Community of Practice that will contribute to sustaining and generalising that change.

## 1. Introduction

A whole-system approach to process improvement in health systems must address the lack of substantial systemic change in quality and safety over many decades [1,2,3]. It is important to analyse the reasons for this imperviousness to change because any solution must have the power and scale to exert leverage over the system in order to foster lasting improvement. The predominant form of intervention in healthcare system improvement is too localised, piecemeal and lacks traction over the forces that maintain system stability. The scale of the problem must be analysed more thoroughly and commensurate solutions must be offered.

The “To Err is Human” report first outlined the extent of patient harm in 2000 [1]. More recently, the Lucian Leape Institute argued in a report on patient safety that the healthcare industry fundamentally needs to redistribute resources from inefficient practices to more efficient ones in order to add value and improve patient outcomes [4]. Yet, it is not enough to critically analyse the reasons for failure in particular cases or, as Dixon-Woods notes, to “admire the problems” in healthcare [5]. Rather, it is necessary to propose a means to directly support action to improve outcomes. The WISH foundation report that shying away from understanding complexity, and instead focusing on safety problems in isolation rather than as a result of many interdependent systems, has led to the failure of the patient safety movement, over the last 20 years, to effect real change [6].

The patient safety movement and the quality improvement (QI) movement in healthcare have been slow to achieve momentum in improving patient, staff and caregiver outcomes. In fact, Braithwaite et al. estimate that in healthcare organisations, nearly two-thirds of initiatives experience implementation failure [7]. Even interventions with proven effectiveness fail to translate into meaningful patient outcomes—what the UK National Health System (NHS) refers to as the “improvement-evaporation effect” [8,9].

The healthcare sector has widely adopted Lean, Six Sigma and Lean Six Sigma as process improvement methodologies, which aim to empower staff to reduce waste by standardising practice [10,11,12,13]. While there are some positive associations between Lean adoption and performance indicators in individual case studies, overall evidence on the success of Lean is mixed [14,15,16]. Considerable time and effort on implementation across the organisation are needed for Lean to be associated with gains in hospital performance, which are in turn mediated by the degree of system maturity, leadership commitment, daily management system use and training [14,16].

The implication of these findings is that a whole-system approach is required, but that this is what has often been lacking. Even where there has been a systemic change, this has taken a considerable time. In order to accelerate the process, the response to the problem must be proportionate to the scale and complexity of the problem. The concept of “Obligation to Act” provides one way of assessing the adequacy of a response to an operational risk. For a response to be adequate, a problem must be seen to be important, the solution must be effective and the pathway to implementing the solution must be viable [17]. Unless these conditions are met, it is unlikely there will be meaningful change. These criteria can be applied to the situation under review in the following way:It may seem obvious that this is an important problem, but the problem must be framed in a way that, in principle, permits some leverage or the possibility of a solution. It is not enough simply to assert its importance. Hence, the analysis must be powerful enough to address the core dimensions of the problem.The solution must not just provide a new analysis of the problem state with aspirations as to how it could be addressed better—it must provide a basis for action, a mechanism that can credibly transform the system. It must generate effective interventions.The solution must work in practice and be effective and usable. It should become embedded in organisational practice and be capable of generalisation to other equivalent contexts.

The core assumption of this paper is that the design of an effective agency of complex and socio-technical system change requires both the understanding of socio-technical systems and the engineering of their development. We describe and evaluate the deployment of a systemic approach to the management of risk, safety, improvement and change that incorporates:The CUBE, a model and methodology for an in-depth analysis of the characteristics of how socio-technical systems and processes function;The Access-Risk-Knowledge (ARK) Platform, a technical system applied with an engineering methodology for intervention, providing leverage to change.

### 1.1. Structure of this Work and Relevance of the Research

In this work, we propose an approach to the management of risk and change in healthcare systems as implemented in a case study on COVID-19 Infection Prevention and Control (IPC). The proposed approach combines elements of socio-technical system analysis (STSA), systems engineering (SE) and a mindful governance model that was developed over several years of research. The ARK Platform was designed and developed with this approach in mind and here we evaluate the initial deployment of the platform across three collaborating healthcare organisations. The overall development and evaluation trajectory for the ARK Platform can be expressed at three levels broadly corresponding to the three criteria outlined above:Defining the problem space as a set of needs to be satisfied. Defining a set of needs for an adequate system in turn sets up a validation exercise: in what way and to what extent does the ARK platform meet a set of high-level operational needs for a system dedicated to managing risk and system change in complex socio-technical systems (STS)?Is the proposed solution effective in practice? Does the framework built into the ARK Platform enable the adequate analysis of a complex operational problem and solution space?Is there a viable pathway to implementation? In this case, the focus is on the architecture and interface design. How well does the ARK Platform support the user in carrying out these tasks?

The background to the research reflects the complexity of the problem at hand, with numerous layers to consider. Firstly, the conceptual nature of the problem space is defined as being amenable to STSA (1.2), which leads to SE as a framework for formulating solutions (1.3). Some core theoretical principles of the CUBE (1.4) and the ARK Platform (1.5) are then briefly described. Next, a set of 16 needs are outlined for a whole-system approach to change; these are subsequently used in the results section of the paper to validate the adequacy of the overall approach to address the problem space (the first question above) (1.6). An operational perspective on the ARK platform incorporating the CUBE then provides an account of how the full model functions in practice (1.7). Finally, the application domain of IPC in the context of the COVID-19 pandemic is briefly discussed as a suitable context for the evaluation of complex system interventions (1.8).

In Section 2, we describe the trial methodology as well as placing the current trial in the context of the broader ARK-Virus project and the development of the ARK Platform over a sequence of implementation phases. Section 3 presents the results of the current trial, which will additionally be used to inform the further development of the platform. Section 4 discusses the results in the context of the ARK-Virus project and of the field. Lastly, Section 5 provides some conclusions to the work and suggestions for future research and development trajectories.

The ARK-Virus project is relevant to the domains of healthcare management, IPC and governance of change in other high-risk industries. It is critical for these industries, and for healthcare in particular, to address the historical failure of change projects to result in demonstrable improvements in outcome. While the ARK Platform is still in its early stages, it offers an approach to risk management that leverages knowledge and information to support change projects throughout the project cycle from design to embedment.

### 1.2. Understanding Socio-Technical System Analysis

In basic system theory, a system is the reality. However, there may be many systems involved in any one reality and it is important to describe the particular system of interest. A model is a representation of how a system works, providing an understanding of cause and effect from a given input to an intended output. The prerequisites to control a system are a goal, a desired system performance and mechanisms influencing the system functionality that are observable and controllable [18].

A key starting point is to describe and define the system. STS are systems in which a relationship between humans and technology exists and for which there is an assumption that this relationship affects the system’s overall performance [19]. Healthcare falls under this definition and has many potential advantages of applying STS theory. For example, adopting a systems approach allows for the identification of “multiple system elements, their interactions and their impact on the quality of care, as well as understanding the key adaptive role of people in the system” [20] (p. 3).

The term STS was coined by Trist and colleagues in the Tavistock Institute in London in the 1950s and later taken up by Klein to recognise the interaction between social and technical factors in organisations [21,22,23]. When trying to implement change, an STS approach stresses the need to consider the technical and social factors and the impact of the change on other aspects of the system [24,25]. STSA involves studying the dynamic interconnectedness of elements of the system at different levels, such as team, processes and information and knowledge.

A well-known STSA framework in healthcare is the Systems Engineering Initiative for Patient Safety (SEIPS), which combines elements of patient safety with Donabedian’s Structure-Process-Outcome model for improving healthcare quality [26]. Donabedian’s model emphasises how three types of information enable an inference of the probable quality of care. Structural measures give consumers a sense of a healthcare provider’s capacity, systems and processes to provide high-quality care. Process measures indicate what a provider does to maintain or improve health, either for healthy people or for those diagnosed with a healthcare condition. Outcome measures reflect the impact of the healthcare service or intervention on the health status of patients. SEIPS is a process model of the healthcare system; it is now on its third iteration [27] and has expanded to include the understanding put forward by Vincent and Amalberti of the patient journey over time and across many interfaces with the healthcare system [28]. SEIPS is a powerful model for mapping out the system as a process with key interfaces and identifying certain types of problems that inhibit a functioning integrated system in healthcare [27].

The SEIPS framework clearly follows system-theoretical models with input and output to and from transformational processes or systems and their interrelations. It supports the understanding of what the system is and some basics of its functionality at a high level. Dawson rhetorically suggests that to have control over change would require perfect knowledge, absolute consensus or power to impose effective control over any extraneous influence; implicitly, this is very unlikely [29]. This poses more than one challenge; one must not only seek to understand such a system, but also to know how one manages, improves and governs the system.

Additionally, healthcare has frequently been called a complex adaptive system (CAS) [2,7], further highlighting the need for a deeper analysis. Often, this label appears as an implicit explanation of why it is difficult to manage, improve and change such a system. Hence Hollnagel et al. call such systems “intractable” and difficult to understand [30]. The three basic terms of CAS provide a good starting point for analysis of the system: the complexity of socio-technical activity, the capacity of such systems to adapt or change in a planned and purposeful way and the consequent capacity to manage and govern in a proactive and integrated manner.

Managing the risks involved in healthcare operations involves moving from diagnosing problems to implementing and embedding solutions. A more comprehensive and systemic analytic process that both generates solutions and guides implementation is needed. Analysis should support identification of the mechanisms that influence the process of change. It is important to delve deeper into the interactions of people, technology and the organisation in order to understand how the normal STS operates, how it might fail and how it might be changed to prevent such a failure.

Research in several consecutive projects has shown that the STSs in development work and change follow the same functional model as the STSs in operations in complex and dynamic contexts, often under varying external conditions [31,32]. Hence, operational and organisational processes and systems benefit from a similar analysis with respect to some core mechanisms of a socio-technical nature. Yet, the process of interest is not only the structure-process-outcome represented in SEIPS, but what will realise and sustain improved outcomes in that model.

### 1.3. STS Engineering

Both the SEIPS framework [27] and resilience engineering [33] lack a systematic method for the process of managing intentional change. Best practice engineering processes have grown into systems engineering (SE) [34], a discipline focussed on how to manage complexity in engineering systems. SE consists of a set of general best practices that may be tailored to a system of interest. Its value lies in a systematic life cycle with a bundle of processes that support the organisation of development work and the delivery and synthesis of multiple system components and their dynamic interconnectedness in functionality. The systemic challenges are similar to that described in the healthcare system.

The essence of the technical processes in SE is the lifecycle of identifying needs, which guides development through formulating a set of requirements that will deliver a functionality. This functionality will have to be demonstrated and validated in its intended context to ensure that it fulfils the originally stated needs. In addition, there are a set of management processes to govern the life cycle process, including management of information, knowledge, risk and measurement. The STS of interest here may benefit from analysis and improvement, as is the case for the work system in healthcare. If the needs are identified based on real and relevant issues and the requirements are based on a thorough understanding of how the system being developed works, then there are conditions for delivering the right understanding, support and tools for change of complex adaptive systems. Our whole-system approach includes the application of engineering principles.

### 1.4. The CUBE—A Socio-Technical Functional Model

Here, we propose our own model to supplement an existing STSA methodology called the CUBE. The CUBE framework has been developed over several years across numerous programmes of research in aviation and healthcare safety [31,32,35,36,37,38] in order to leverage STSA in support of organisational change. The framework was designed to support an analysis that could span a set of diverse theoretical approaches and perspectives. The CUBE allows for a rich understanding of the system to be built around four domains:Sensemaking: incorporates Weick’s work on how individuals operating within the system make sense of it, often through practical action [39]Culture: incorporates Schein’s [40,41] and Pigeon and O’Leary’s [42] work on cultureSystem: incorporates Perrow’s functional focus on complexity and coupling [43] and accounts for both formal and informal elements, i.e., Policies, Procedures, Protocols and Guidelines (PPPGs), as well as the sequence of activities that normally takes place [38]Action: incorporates the flows of information, knowledge and understanding and anything that happens in the system that is recordable or measurable [44]; this can be analysed at different levels such as individual actions, team performance against a standard, activity sequences, or key outcome, process and balancing measures in relation to system performance [45]

These may seem to be alternative approaches but are in fact fundamentally mutually compatible and complementary theoretical frameworks for a whole-system approach. Each in its own right has weaknesses with respect to a whole-system approach:Cognitive science has a local focus, which is not systemic (in the socio-technical sense)Organisation theory does not, in general, address functional aspects of processes or the achievement of valueProcess theory (which does address the value stream) does not effectively address the role of peopleKnowledge Management theory lacks the analysis of the content of knowledge that is held and partially shared by agents in the system

Therefore, the attempted synthesis is necessary, because otherwise it is impossible to disentangle the theoretical stance of the investigator from the substance of the case or issue being investigated. If this is the case, then it vitiates any possibility of a scientifically neutral account of the dynamics of STSs including the processes of implementation and change. Evidence acquired with one or the other approach may overlook significant understandings that may be observed from another perspective.

### 1.5. The ARK Platform—A Technical and Analytic Support for Driving Change

Even an adequate basis of analysis is insufficient for driving change. Analysis must be supplemented by a mechanism to put the conclusions into practice, to do so again and again and to have some oversight over the whole process. This implies a practical concept of governance with tools and methods to support it. From High Reliability Organisations [46] through Resilience Engineering [33] to Safety II [30], there is a long evolution of concepts of governance that have struggled to find a substantial practical application.

The CUBE methodology starts with needs or a problem formulation, then supports the development of a concept or solution, integrates solutions through planning and preparation, implements designs in operations (work system) and validates the actual outcome. As with most SE, this is an iterative process, rather than sequential. For example, in the design stage, a solution may be unfeasible due to lack of resources, and hence, the team must backtrack and discuss priorities in terms of needs or requirements for change. Each stage of the lifecycle evaluates the conditions for achieving a certain outcome and the mechanism that delivered the outcome. The CUBE methodology supports a systematic approach but also ensures that the process of change is monitored and documented to allow follow-up and learning. In the ARK Platform, this is supplemented by the Context-Mechanism-Outcome model [47]. By documenting how an analysis and interventions are performed, it becomes possible to return iteratively if a later stage does not fulfil needs or requirements.

The concept of Mindful Governance of Operational Risk [17] was developed as a way of operationalising a conceptual approach to mindful organising put forward by Weick [39]. Weick’s approach argued for a set of general dispositions of individuals and collectively of organisations (e.g., preoccupation with failure, reluctance to simplify, and sensitivity to operations). However, for “mindful organising” to occur, there needs to be an actual flow, transformation and management of information, not just the right mindset. Therefore, the development of the initial concept of Mindful Governance was accompanied by the development of practical tools for gathering narratives from operational staff and demonstrating methods for analysing complex patterns of operational data [17,48]. Subsequently, the development focus moved towards implementation of change and improvement following the assessment of risk [49,50]. This also consolidated the realisation that the accumulated data from much organisational activity provided a core mechanism for engineering change—knowledge of what has happened in the past, engineered to support the performance of those complex organisational tasks in the present. The accumulation of knowledge depends on the application of STSA and risk assessment to all phases of activity. This seeks to provide an advance on the current state of the art. However, the more substantial innovation will be to leverage the accumulated knowledge to improve the functioning of the target system as a whole.

In the current paper, we report on the first full trial implementation of the prototype platform designed to support the development of an effective mechanism for effecting system change.

### 1.6. Elements of a Fully Systemic Approach to Change: Needs Gathering

In this section, we outline a set of fifteen minimal needs of an approach to healthcare change management that is fully systemic in nature. Needs were identified based on the combined expertise of the group and the available evidence on STS governance and change (seven needs), healthcare quality improvement (four needs), risk and safety governance (two needs) and data governance (two needs). The needs identified are the basis for further development of the ARK-platform to support the management of system complexity, adaptation and change, and system governance. Later in the paper, we evaluate the ARK Platform against these needs to assess how fully the current version of the platform works as a solution to the issue of change in healthcare.

#### 1.6.1. STS Governance and Change

One of the few attempts to put forward a whole-systems account of successful system change was a longitudinal cross-industry comparison of common factors that underlay long-term competitive success [51]. This study identified five factors, which concerned scanning the competitive environment, linking operational change to competitive requirements, treating people as assets rather than cost, leadership at all levels and ensuring coherence across complex strategic initiatives. This provides one reference point for deriving a set of practical requirements for a systemic approach to healthcare improvement. Mesjasz sets out five dimensions of complexity that provide a starting point for understanding STS activity: multiplicity of cause and effect, non-linearity, emergence, human agency and self-organisation [52]. Based broadly on these two sources, a set of seven generic requirements are put forward for the effective governance of a functional STS, such a system having a set of goals, objectives, multiple stakeholders, and complex operational processes with a range of outcomes. These form an initial set of needs to address to overcome the challenges of system improvement in healthcare.
Multiple interacting causes and consequences. Understanding the multiple interacting causes and consequences of specific operations requires diverse data sources that meaningfully represent process activity, quality and outcomes.Non-linear relationships. An adequate theoretical understanding of STS states and transitions is needed. This includes the strategic role that operational change can play or the potentially transformative role of new knowledge or information.The role of people. The role of people as active agents in the system and assets to system change needs to be factored in.Self-organising tendencies of adaptation and change. In relation to the self-organising tendencies of adaptation and change, it is necessary to overcome the false dichotomy in theories of change between emergence (change is spontaneous) and managerialism (change is controlled) [53]. To achieve this, it is necessary to have a common framework to understand the change process itself together with the productive role of people and leadership at all levels.Adequate basis for action. For Pettigrew and Whipp, the Strategic Environmental scan creates a rationale for action at the highest level and requires looking beyond the organisation to the level of the whole industry or services to the community [51]. However, a basis for action needs to be understood at all levels—what makes something important, a solution effective, and an implementation pathway viable?Emergence. The identification of emergent, systemic factors requires multiple case studies that adequately record complex activity. Aggregating such a whole-system perspective requires building a dynamic synthesis of system activity across an organisation (and often beyond its boundaries) to enable the evaluation of the strengths and weaknesses of system activity.Strategic coherence. Coherence across multiple initiatives requires a rich knowledge resource for synthesising the understanding of activity across the system. Such a knowledge resource is arguably the starting point for the building of strategy and policy based on accumulated evidence.

#### 1.6.2. Quality Improvement and Lean Programmes in Healthcare

The empirical evidence of improvement and change shows that sustaining improvement is more the exception than the norm. Where it works, it takes a long time and much organisational effort to embed change capability. We identified four needs from a QI perspective to support a full systems approach to change management, thus improving the quality of implementation and expediting the process.
8.Training and education. Staff competence cannot be assumed. McNicholas et al. evaluated multiple Plan-do-study-act (PDSA) cycles to identify how to provide improvement support [54]. They found that the methodology was rarely implemented properly, although there was some evidence of an improvement over time. The complexity of QI was underestimated, and training and support were insufficient. Savage et al. identified two contrasting styles of medical leadership—a virtuous cycle of management through medicine (an ability to see the wider picture, beyond one’s own domain and hence an ability to see strategic challenges at the system level) and a vicious cycle of medical protectionism (motivated by clinical identity and professional objectives) [55]. In the virtuous cycle, willing leaders continually improve their own management and leadership competencies, deploying participatory practices that cultivate medical engagement amongst staff.9.Improvement processes embedded within normal management activity. The capability to develop and foster an ongoing learning framework to sustain the improvement and to generalise the approach to other parts of the system is often missing from even successful Lean improvement initiatives [56]. The implication is that such an improvement framework needs to become embedded in the normal management processes of the organisation, otherwise there is no mechanism to plan improvement strategically nor to consolidate the lessons from individual initiatives [14,16].10.Provide a systemic methodology for collecting evidence. Dixon-Woods argued that there needs to be a sounder evidence base for QI interventions [57]. Some interventions that she noted are probably just not worth the effort and cost (e.g., having nurses wear “do not disturb” tabards during drug rounds) and some QI efforts, perversely, may cause harm (e.g., when a multicomponent intervention was found to be associated with an increase rather than a decrease in surgical site infections). QI methods need to both sustain the organisational processes that enable successful improvement initiatives and, at the same time, gather and analyse the data around those processes in a way that produces case studies that meet scientific criteria.11.Produce shareable knowledge within and between organisations. A broad evidence base consisting of multiple case studies will lead to standards of good practice that support timely and effective decisions. Each case study leads to a broadening and deepening of that evidence base.

#### 1.6.3. Risk and Safety Governance

The challenges of implementing safety and risk management systems and integrating these with Lean initiatives have been studied in the aviation industry, with clear implications for similar programmes in healthcare. Since 2014, the aviation industry has had a regulatory requirement for a safety management system (SMS), which applies a process-based approach for safety management originating from the field of quality management. This has led to a change in aviation from an outcome-based to a performance-based regulatory assessment, moving away from a reactive manner of counting the number of events and incidents and instead requiring demonstrated system controls as a measure of safety or safety performance [58]. To demonstrate safety performance means to have the capability to be responsive to identified hazards, thus controlling or reducing operational risk.
12.Common organisational capabilities for development, improvement and change. In safety management and risk management, the focus has traditionally been on the earlier part of the lifecycle of identifying hazards and analysing risk, such as monitoring system status and suggesting mitigations and recommending improvements. The performance-based approach requires a dual function: identifying hazards, as well as monitoring and influencing change and providing evidence of actual improved system outcomes linked to the change initiative. Depending on organisational capability, system improvement is assessed on a maturity scale from compliance to excellence. Organisational capabilities for development work or improvement follow basic STS principles known to support actual change, such as participation, leadership, communication and structures for the flow of information.13.An integrated approach is needed between safety and risk and Lean. Findings from a continuous research collaboration with a Lean airline identified challenges with implementing a SMS [32]. Recent interviews and a SMS maturity assessment found that the challenges remain after 15 years of compliance [59]. The studied airline implemented Lean in parallel to their first version of a SMS. However, the two management systems do not join-up. One reason is identified to be the lack of organisational capability that Lean is assumed to provide. Another is that safety is organised from a smaller support function, with the intention of protecting the importance of safety. This leads to isolation, which instead marginalises the impact of safety activities.

#### 1.6.4. Data Governance

In recent years, “big data”, “data science” and “data analytics” have tended to dominate the discussion as the key enablers for digital transformation of organisations, such as enabling a whole-system approach to process improvement [60]. However, in parallel with this technology-centric pathway, there has been a rising awareness of the importance of developing a strong data governance infrastructure at both the organisational and technological levels [61]. Data Governance can be defined as “the exercise of authority and control (planning, monitoring and enforcement) over the management of data assets” [62]. When data or evidence are needed to analyse the problem space in an organisational context, it is necessary that the data are trusted, are of appropriate quality and can be efficiently located and processed using common terminologies or efficient conversion techniques [63]. Data governance practices and methods such as use of data catalogues and data quality processes, the combination of operational data experts with IT systems and support, and the generating of governance-oriented metadata for monitoring [62] and control will support system change.
14.Data governance infrastructure to support system change. Technical barriers to deploying whole system change management include siloed information systems for different hospitals and care providers [64]. One established technical solution for enabling data portability, interoperability and fusion is the use of formal engineering ontologies based on computer science knowledge [65]. Ontologies can be used to progressively integrate complex datasets in dataspaces that provide basic services such as search and access control while delaying the costs of full data integration [66]. For a multi-organisation, multi-site or multi-stakeholder analysis, it is necessary to support data federation in order to enable collaboration, evidence pooling for rare events and evidence-based best practice recommendations [67]. In addition, the introduction of stronger data regulation such as GDPR has meant that organisations now have compliance requirements for their data handling. Unfortunately, many organisations have inflexible, defensive privacy and data protection strategies focused on auditing rather than risk management, transparency and accountability. Thus, a new generation of privacy-aware data processing systems are required [68].15.Privacy-aware data federation. The ARK platform seeks to develop an evidence-driven capability for Enterprise Risk Management [69] in which the addressing of diverse sources of risk in a commensurable manner based on a unified knowledge base enables the addressing of the value dimensions of multiple stakeholders, and hence, the realisation of the core principles of value-based healthcare [70].

### 1.7. Applying the CUBE in Practice: The ARK Platform

Operationally, the CUBE consists of a 96-item questionnaire that guides safety experts in identifying, assessing and classifying risks, as well as in planning, executing and evaluating risk mitigation actions. As described in the Introduction, the CUBE is built around four domains: System, Action, Sensemaking and Culture. The CUBE is further divided into four system aspects: Goals, Process, Social Relations, and Information & Knowledge. Figure 1 shows the sixteen dimensions of the CUBE framework across the five project phases.

Through our previous work on the CUBE, it became apparent that an integrated platform was needed for in-vivo deployment. Thus, the ARK Platform [49] was created to expand the CUBE’s practical capabilities and to provide a way to embed the CUBE approach to risk management within the organisation. The ARK Platform supports a fuller analysis of risk and a complete mitigation of that risk. The goal is to design a system that will have a whole-system approach to process improvement in health systems and, in particular, to address complexity, adaptation and governance. The CUBE provides a model of how the STS functions and the ARK Platform was designed to address critical gaps in the governance of complex STS change. It was also recognised that it is not enough to support a better analysis of risk and mitigation processes in particular cases; it is also critical to develop and synthesise knowledge from a combination of many cases in order to build an evidence base of effective practice. These gaps lay the ground for identifying needs that the ARK platform is to fulfil.

The ARK platform uses the CUBE framework to leverage information in support of change. By completing a project on the ARK platform, users use the CUBE to build a model of how to manage risk and change within a complex STS. The result of an ARK project is a well-supported analysis of a full change cycle that allows for cross-project comparison. It contributes to the shared evidence on change management and to an organisational process of learning to improve the quality of the cycle as a whole.

The platform is pragmatically arranged around project phases: problem, solution, plan and prepare, implement, and verify and embed. Progress through the phases is illustrated by the evolution of risk (Figure 2). An initial risk assessment can be imported to the ARK platform from an external risk register. That assessment will become further assessed and amended using the CUBE and other associated evidence. The solution stage supports a derivation of that risk—the Projected Risk, given the implementation of the solution. The Plan and Implementation stages introduce a new risk—the Risk-in-Change (RiC)—that is projected at the plan stage and actual at the implementation stage. Finally, the Residual Risk is the actual risk remaining after implementation. It should, insofar as these have been accurately estimated and validated with associated evidence, approximate the combined outcome of the projected risk at the solution stage and the risk in change during implementation.

Each project stage focuses on deciphering what needs to be changed at that phase of the project implementation sequence and on generating leverage over the system in order to achieve the change. At each stage, users complete a Project Analysis, CUBE Analysis and Risk Assessment. The Project Analysis, which includes an initial risk rating and CMO analysis, acts as an interface between the CUBE Analysis and the Risk Assessment. In particular, the CMO provides a rationale for the Risk Assessment and justifies the focus of the project. Rather than merely summarising, the CMO provides a synthesis of the information in the CUBE Analysis in a way that leads into a more detailed STSA. Users are additionally able to upload evidence from various data sources to the Project Analysis page, with further evidence-interlinking capabilities under development. Uploaded evidence can be shared across the user organisations to develop the shared knowledge base or can be kept within the organisation or team. Figure 3 shows the elements of the ARK platform and the purpose of each.

The ARK Platform (Figure 4) is used to build and maintain a unified knowledge graph of risks and projects that links available datasets on practices, risks and evidence. Knowledge graphs [71] have recently received a lot of attention as a way to share insights from machine learning or predictive analytics processes. They are a well-known method to support data integration [65] and a popular format for meta-data definition [72]. At the technical centre of the ARK platform is a unified model of risk (expressed as a knowledge graph) that bridges traditional qualitative risk evidence and quantitative operational or analytics data. This makes large-scale evidence collection and risk analysis more tractable by transforming human-oriented quantitative risk information into structured, machine-readable data suitable for automated analysis, querying and reasoning. This is achieved using the ARK Mindful Governance of operational risk formal ontology based on logical semantic models and enabling safety analyst-oriented text fields to be annotated with a domain taxonomy ontology based on safety, healthcare and other domain-specific concepts. This makes even highly specialised socio-technical risk analysis textual data amenable to machine processing.

ARK’s combination of the mindful risk governance methodology with metadata-driven data governance will enable proactive evidence dataset recommendation (unknown in the state of the art, which still relies on search [73]), knowledge extraction from semi-/un-structured risk reports and combination with structured operational data to create a new, unified risk evidence base that is unknown in existing, highly siloed safety systems that emphasise manual risk analysis. The combination of these features with a rich modular risk and governance ontology supporting classification and reasoning provides an advanced framework for targeted data collection from operational staff, risk assessment, governing evidence-based organisational change for change project monitoring, risk information distillation and automated risk information circulation and feedback. When these processes are in place over many system-change projects, it will be possible to conduct semi-automated multi-project analysis and the distillation of best practices into shareable, privacy-aware knowledge bases based on Linked Data. The development of this evidence base is critical to optimise the effectiveness of processes.

The ARK data governance services support integrating siloed healthcare risk datasets, interlinking local knowledge to web-based sources, providing structured metadata about evidence, federating sensitive data from multiple organisations and enforcing privacy when converting local sensitive data to sharable evidence. Deliberative human control of safety analysis and recommendation is at the heart of the Mindful Governance methodology and the ARK platform. Nonetheless ARK aims at saving safety experts time and effort with data aggregation, classification, ranking and structured Mindful Governance workflows.

The ARK-Virus project, introduced next, aims to identify the most effective analysis and automation steps that are possible, such as: dataset recommender systems, evidence visualisation, time series data annotation, natural language processing-based annotation and data analytics integration.

### 1.8. Infection Prevention and Control as a Complex System

The ARK-virus project provided the opportunity to test and evaluate the platform in the context of a complex healthcare problem: the risk management of IPC. In this section, we outline some of the complexities of IPC in a hospital context, which justifies the contention that this is a complex system.

Nosocomial infections, otherwise known as healthcare-associated infections (HCAIs or HAIs), are those infections acquired in a hospital or healthcare service unit that first appear 48 h or more after hospital admission or within 30 days after discharge following inpatient care [74]. They are unrelated to the original illness that brings patients to the hospital and are neither present nor incubating at the time of admission. A systematic review using data up to 31 March 2020 estimated that 44% of COVID-19 cases were nosocomial—acquired in hospital by patients who were admitted for other reasons. In previous SARS and MERS pandemics, 33% and 56% of all diagnosed cases were nosocomial [75]. This COVID-19 research reports on cases early in the pandemic as guidelines were still evolving.

Adults who acquire COVID-19 whilst already hospitalised are at greater risk of mortality compared to patients admitted following community-acquired infection; this finding is largely driven by a substantially increased risk of death in individuals with malignancy or who had undergone transplantation [76].

The acquisition and transmission of nosocomial infections is a complex STS problem with many elements. Many patients are frail and immunocompromised at the time of admission, are routinely accommodated in communal wards with shared equipment and facilities that are often not designed or maintained sufficiently and are exposed to complex treatments, interventions and devices that can further compromise their natural barriers (i.e., skin or immunity). In addition, patients are cared for by healthcare professionals who themselves can be a source of nosocomial transmission as they move from patient to patient. These risks are further heightened by the high-volume use of anti-microbial agents, which creates selection pressure for the emergence of resistant strains of microorganisms.

Accordingly, because of this complexity, taking a whole-system approach to the risk management of IPC in one hospital, for example, would require looking at numerous sources of evidence, e.g., IPC governance arrangements and reporting, HCAI surveillance data, environmental hygiene assessment findings, antimicrobial use audits, invasive device monitoring, hand hygiene training and practice audits, staff training, relevant clinical audits, adverse incidents reporting, risk registers and patient feedback.

Evidence on the effect of COVID-19 on other HCAI rates is emerging now and this further highlights the complexity of the problem. In 2020, in the US, significant increases were found compared to 2019 in four serious infection types: central line-associated bloodstream infections, catheter-associated urinary tract infections, ventilator-associated events and antibiotic-resistant staph infections. The largest increases were bloodstream infections associated with central line catheters that are inserted into large blood vessels to provide medication and other fluids over long periods. Rates of central line infections were 46 to 47% higher in the third and fourth quarters of 2020 compared to 2019 [77]. The data are based on the US’s largest HCAI surveillance system, the National Healthcare Safety Network. With dramatic increases in the frequency and duration of ventilator use, rates of ventilator-associated infections increased by 45% in the fourth quarter of 2020 compared to 2019 [78].

Prior to the pandemic, a widespread decrease in HCAI incidence had been observed across US hospitals, making the witnessed increases in HCAI particularly concerning. In 2020, increased patient caseload, staffing challenges and other operational changes limited the implementation and effectiveness of standard IPC practices. IPC practices had to adapt to worldwide Personal Protective Equipment (PPE) shortages and deal with fear of infection among healthcare workers [78].

The study found that two other types of HCAIs remained steady or declined during COVID-19. Surgical-site infection rates did not increase as fewer elective surgeries were performed and those that were undertaken occurred in operating rooms with uninterrupted IPC processes that were separate from COVID-19 wards. In addition, no increase was found in *Clostridioides Difficile*, or *C. diff*, a serious bacterial infection that occurs after antibiotic use. The study said lower rates of *C. diff* may be a result of increased focus on hand hygiene, environmental cleaning, patient isolation and increased attention to the use of PPE.

## 2. Materials and Methods

The case study methodology was developed along two parallel objectives. The first was a user evaluation of the ARK platform and validation of the extent to which its functionalities were utilised during the study. The second was to assess the extent to which the ARK platform meets the high-level operational needs for a whole-system approach to risk management in complex STS (outlined in the Introduction). To fulfil these objectives, two research design approaches were combined—user evaluation of technology in use [79] and action research [80,81]—using the ARK Platform to establish projects in a relevant domain within each collaborating organisation. In line with action research principles, methods were continuously refined throughout the course of the study in response to input from the collaborators.

### 2.1. Use Case: The ARK-Virus Project

The ARK-Virus project is a collaboration between an academic team and a Community of Practice (CoP), which includes quality and safety staff from a large 1000-bed urban academic teaching hospital, medical staff from a private renal dialysis service, and management staff from a large urban fire and emergency medical services (EMS) provider. The ARK-Virus project was designed to develop the ARK platform through the initiation and management of an IPC project in each of the three organisations. The development of these organisational projects was in line with the Sigtuna principles, which provide criteria for the design, implementation and evaluation of specific interventions, e.g., engagement of key stakeholders, alignment with organisational objectives, working with existing practices, developing organisational learning and evaluation, and transferring knowledge beyond the organisation [82]. The group of participating organisations was explicitly established as a CoP to share knowledge and experience in a way that would foster improved practice and contribute to best practice standards. Their diverse roles within the health system were seen to be an advantage in this.

PPE is a critical component of IPC and, as such, up to date and situationally aware risk management is critical to ensuring that PPE guidelines are understood, implemented and maintained. This is particularly relevant during the COVID-19 pandemic where it is vitally important to monitor PPE use to optimise its effectiveness in reducing the risk of virus transmission in healthcare settings. A key focus of the ARK-Virus Project is to develop a shared evidence base of PPE compliance data across the participating organisations. Using these data, the CoP can then conduct socio-technical risk analyses, via the ARK Platform, using the CUBE methodology. Effective COVID-19 IPC risk governance requires engagement from many actors across the entire healthcare organisation. This is also facilitated by the CUBE Mindful Governance methodology, which uses risk-oriented, multi-actor, safety improvement projects as its core unit of work. Using the ARK Platform, users can link these safety improvement projects to supporting evidence such as datasets on IPC best practices, COVID-19 transmission data and organisational IPC/PPE data. By putting the ARK Platform in place over many IPC projects across multiple organisations, it will be possible to collate the resulting best practice data into a shareable, privacy-aware, linked knowledge base. Development of this integrated evidence base is critical in optimising the effectiveness of PPE and in understanding the factors that influence PPE compliance.

The ARK Virus Project has four development trials planned, which are outlined in Table 1 below along with their associated research activities. The results presented in this work were collected at the end of Trial 1 and will be used to enhance the platform in preparation for Trial 2.

### 2.2. Training in Advanced Risk Knowledge

Prior to and throughout Trial 1, collaborators were provided with a combination of pre-recorded and live online trainings on how to apply the CUBE methodology and how to use the ARK Platform. This training was developed in collaboration with the Operational Risk: Implementing Open Norms (ORION) project, a 32-month Erasmus+-funded project that designed, developed, delivered and evaluated a training programme to support and advance the implementation of SMS in safety critical sectors including emergency services, healthcare and aviation [83]. This course was developed around a concept of Advanced Risk Management and supported by the ARK platform. Within an operational management framework, it aims to build the capability to analyse the risk in complex problems, to use and analyse diverse sets of data, to manage the risk in implementing solutions and, through, all this to build a strategic capability to manage system risk.

The ORION programme included five training modules:Operational risk and organisational hazards.Proactive risk management—formulating a project.Operational risk model—managing advanced data analytics.Measurement and monitoring for safety assurance.Implementing system change—managing a project.

The first module included basic organisational capabilities derived from STS and engineering principles. The second introduced the CUBE and initiated the discussion of setting up systemically resourced projects. The third supported the management of advanced data analytics. The fourth discussed the joint performance management required in a joint-up management combining Lean and risk management measurement and monitoring. The last module consolidated a whole-system approach linking change and strategy. The modules were provided as a web-based course with live sessions to discuss and initiate a project such as ARK-Virus. Module 2, with a focus on initiating a project, was delivered with on-line workshops to support Trial 1. Further modules will be delivered to support the next set of trials.

### 2.3. Evaluation Strategy

The version of the ARK Platform that was evaluated as part of this trial was developed during Trials 0 and 1 of the ARK Virus Project. This was the first version of the platform to be extended for the healthcare domain and it was still in the early stages of development when evaluated. A goal of the trial was to collate a set of requirements to improve and refine the platform for Trial 2 of the ARK Virus Project. Evaluation of the platform was conducted in two parts: a user evaluation and an evaluation against the requirements.

From June to August 2021, collaborators were given access to the platform and asked to initiate a risk management project relating to COVID-19 IPC. Collaborators aimed to complete the Problem and Solution Stages of their chosen project with the view that the remaining stages would be the focus of future trials. Over the course of the trial, each of the organisations adapted the platform to suit their needs, generating more general models of risk management of IPC. Members of the research team were available to liaise with the organisations and provide technical support as needed. Each month, a plenary meeting was held in which users had the opportunity to ask questions, deliver feedback, update on the status of the project, and discuss their experiences with each other and the research team.

At the end of the three-month trial, platform users (*n* = 7) were invited to take part in an anonymous follow-up survey (see Appendix A) and an online focus group (Appendix B). The aim was to evaluate user experiences from a technical and an operational point of view to inform future iterations of the ARK Platform. Six of the seven users took part in the online survey and seven took part in the focus group. Each of the three collaborating organisations was represented in both parts of the evaluation. The user evaluation was supplemented by ARK Platform Trial Metrics. All research ethics principles were adhered to including timely informed consent.

The anonymous survey was administered using Google Forms and used to collate feedback on participants’ experiences of using the platform. The questionnaire included two sections:User-Interface Feedback.ARK Platform Feedback.

The User-Interface Feedback portion of the questionnaire evaluated the ARK Platform’s user-interface via the System Usability Scale (SUS) [79] and four open questions exploring what users liked and disliked about the platform, suggested areas for improvement and other suggestions for the user-interface. The SUS is a highly robust tool used to measure system usability. The SUS consists of 10 usability statements, both positive and negative, about which users rate their level of agreement on a five-point Likert scale ranging from Strongly Disagree to Strongly Agree. This produces a baseline usability score, which can be used to make future iterations of the platform more user-friendly.

The ARK Platform Feedback section of the questionnaire included ten questions on general experiences using the ARK platform. Seven of these were open questions and five required participants to rate the usefulness of different ARK Platform features on a five-point Likert scale from “not at all useful” to “very useful”. The results of the SUS were calculated as per the tool’s instructions and simple quantitative analysis, based on the most common responses, was performed on all other closed-ended questions. Open-ended questions were analysed for common themes, and quotes that were relevant to the goals of the research were extracted.

In the focus group, participants were asked to discuss a series of questions relating to their experiences using the platform and the CUBE methodology (see Appendix B). Topics centred around what participants liked or disliked about the ARK Platform and the CUBE, as well as the usability and utility of both. Subsequently, they were asked to discuss the impact of the ARK-Virus project on their organisation and operational requirements moving forward. Questions were informed by the literature and informal discussions with collaborators throughout the course of the trial. Following the focus group, recordings were transcribed, and thematic analysis was conducted independently by two members of the research team. Preliminary results were presented to the users in a follow-up meeting, giving them the opportunity to make corrections or additions. No changes to the themes were made as a result of this meeting, but the findings were discussed in further detail amongst the collaborating organisations.

In part two, the ARK Platform was evaluated against the set of needs for a systemic approach to change outlined in the Introduction. The research team, informed by the trial results and the literature, compiled a list of features of the ARK Platform relating to each of the identified needs. The evaluation according to needs was conducted by the research team with input from the CoP users and incorporated evidence from the survey and focus group. There were two evaluation criteria: whether the need was addressed in a feature of the platform that had been developed (verification) and whether that feature was used and evaluated in the trial (validation).

## 3. Results

The COVID-19 pandemic has highlighted the importance of developing organisational capacity for rapid change in response to emerging information, as well as the sharing of information across organisations and sectors. Users reported that both IPC and PPE compliance projects seemed to be well-supported by the ARK Platform in these areas. However, the ARK Platform’s utility was not limited to COVID-19 IPC projects. In fact, across the participating organisations, three different projects emerged. One focused on risk management in general, one focused on risk management of IPC in general, and one focused on the use of PPE over the course of the pandemic. As the ARK-Virus project continues, users are interested in seeing the capability of the platform to put forward a general model for IPC risk management that could be easily applied to future infectious disease threats.

In this trial, users were asked to focus on completing the Problem and Solution stages of the ARK Platform. Three risk governance projects, one by each participating organisation, were created over the course of the ARK Platform Trial. Two of the organisations completed all of the Problem Stage and nearly all of the Solution Stage, while the third organisation partially completed the Problem Stage. Two of the three organisations linked evidence to their risk analysis. The delay in completing the Solution Stage was, in part, due to work demands during the ongoing COVID-19 pandemic as well as administrative delays in signing legal documentation such as data processing agreements and non-disclosure agreements. Users were also asked to keep track of missing features, bugs and ideas for improvement throughout the trial. They could send feedback to the research team at any time ahead of the end of the trial period.

At the end of the trial, users provided further feedback through an anonymous questionnaire and focus group discussion. The results of the questionnaire and discussion were analysed to answer our research questions “Does the framework built into the ARK Platform enable the adequate analysis of a complex operational problem and solution space?” and “How well does the ARK Platform interface support the user in carrying out these tasks?” Input from the CoP users and the project research team was used to answer the third question, “In what way and to what extent does the ARK platform meet a set of high-level operational needs for a system dedicated to managing risk in complex socio-technical systems (STS)?”

Users gave the ARK Platform a SUS score of 47.5 out of 100, indicating significant issues with usability. They highlighted a few bugs and drawbacks, as well as several elements they liked; however, the feedback was most focused on potential additions to the platform. The results were more favourable regarding the usefulness of the platform, with users generally reporting that it was useful for their organisation and that they could envision it being even more useful in the future. In terms of the focus group, users identified five benefits and two drawbacks of the current version of the ARK Platform. This led to a good discussion of the trajectory of the platform and its potential applications in practice. The evaluation of need fulfilment found that six needs had a feature of the platform developed and validated, five had a feature developed but had not been validated, and four do not yet have a feature developed.

### 3.1. ARK Platform Feedback Questionnaire

As mentioned, the ARK Platform Feedback Questionnaire included two main sections—User-Interface Feedback and ARK Platform Feedback.

#### 3.1.1. User-Interface Feedback

The User-Interface Feedback section of the questionnaire focused specifically on collating feedback on the ARK Platform’s user-interface. This section consisted of the SUS and four open questions exploring what users liked and disliked about the user-interface.

The overall average SUS score across the six participants was 47.5 out of a possible 100. As the average SUS score was 68, a score of 47.5 was below average. Figure 5 shows how ARK Platform SUS score stands within the percentile rank and letter grades associated with the SUS. Here it can be seen that a score of 47.5 was below the 10th percentile and received an F grade. This indicates that there were significant usability issues identified, which need to be addressed on the user-interface. As the platform was still in Stage 1 of development when evaluated, a low usability score was to be expected. The SUS combined with user feedback generated rich results, which will be used to inform the further stages of development.

With respect to what participants liked about the platform’s user-interface, they mentioned the progress bar visualisation, the CUBE summary visualisation, “the ability to move through the five project stages and the project analysis/cube analysis/summary/risk assessment on the one interface” and how the interface “guides the user through a systematic approach”.

In terms of dislikes, participants mentioned issues with saving inputted data, including both slow saving times and saving bugs, as well as the fact that there was no data output/report generation feature available. One participant stated that they found the user-interface to be “slightly disorienting”. Another mentioned that they felt the CUBE had “too many sub-questions”; however, this was not so much a user-interface issue as it was a process issue.

When asked what key feature/improvement they would like to add to the ARK Platform, user-interface participants mentioned:An ability to see any changes made since the last login;Explanations of the key terms used in the questions;Project reports/outputs;Better navigational support;Additional progress information.

Finally, when asked whether they had any other feedback on the ARK Platform user-interface, participants suggested integrating training material into the platform, improving the processing speed of the user-interface and improving system navigation. One user commented that there were too many CUBE sub-questions and that “the questions were also difficult to ground into the reality at times”; however, this was again not a user-interface issue but a process issue.

Using the above user feedback and the results of the SUS, specific usability issues will be targeted and improved for future versions of the platform.

#### 3.1.2. ARK Platform Feedback

The ARK Platform Feedback portion of the questionnaire consisted of 10 questions—seven open questions and five closed questions. The closed questions investigated the usefulness of the ARK Platform and the usefulness of its individual components. The results to these questions are summarised in Table 2, which shows that users generally found ARK to be useful.

When asked what their overall experience of using the ARK Platform was, one user reported that “using the platform supported us meeting as a group and discussing a big issue in the organisation” and that this “allowed for analysis of the problem from multiple perspectives”, which supported building “a rich picture from different managers of the issues.” Users also reported that the platform was initially “daunting”, and that it “took some time to become familiar with the interface”. In addition, users stated that the risk analysis process was “more time-consuming than expected”—this could be linked to the high number of CUBE questions that users reported in the User-Interface Feedback portion of the questionnaire.

When asked about the impact of the ARK Platform on the risk management of IPC in their organisations, users mostly reported that it was too early to say at this stage of the ARK-Virus Project. However, users did report that they could see the benefits of:Different stakeholders being able to use the platform to build a “collective solution”;“Developing a repository over time of risks, controls, mitigations that work”;“Learning from other organisations about how they dealt with risks and managed the implementation of mitigants and recommendations”;“A more integrated approach to the prevention and control of healthcare associated infections”;Discussing “domains like social goals, processes and culture that are rarely explicitly documented or discussed when looking at a patient safety risk”.

When asked the type of risk analysis that users performed using the platform, responses included:The Irish Health Services Executive Impact Cause Context (IxCxC) approach;Compliance with IPC measures;Internal and external risk assessments.

Finally, when asked whether they had any other feedback on their experience of using the ARK Platform, one user reported that the platform “provides a useful structured approach to managing risk” and another reported that they “would have benefited from a project with a smaller scope to familiarise ourselves with the platform”.

It can be seen from the above results that users consider the ARK Platform to be useful and to have great potential for healthcare risk governance. As with the user-interface feedback, the feedback received from this portion of the questionnaire will also be used to further develop the ARK Platform in preparation for Stage 2 of the ARK Virus Project.

### 3.2. Focus Group Feedback

Feedback from the workshop mirrored the findings of the survey, with users highlighting five benefits: expanding risk management, supporting transparency, building evidence, engaging stakeholders and sharing knowledge. Overall feedback was positive regarding the platform’s impact and potential, but there were a few shortcomings discussed as well. In addition to issues with the technical usability of the platform, users flagged two drawbacks: unclear workflows and lack of outputs.

Benefit 1: Expands Risk Management. Participants felt that interacting with the ARK Platform promoted a broader awareness of risk management and implementing change by forcing users to consider the social aspects of change, culture, sense-making and communications. Users referenced the fact that implementation of a PPPG does not necessarily mean it has been enacted. There is a need to factor in how people interpret that PPPG, how it is understood or misunderstood, and whether it was enacted as intended. The approach also provides an integration pathway for combining risk assessment and improvement projects, which typically occur in different spaces within the organisation, in order to give what one participant describes as a “very rich governing picture”. One organisation found that this perspective made the platform compatible with their journey towards an Enterprise Risk Management approach and complemented existing risk assessment processes.

Benefit 2: Supports Transparency. The ARK Platform supports transparency by assigning responsibility for project roles and actions and by tracking the resources used throughout a change project from the problem stage to the verification stage. As in the survey, users were interested in ways to improve this element of the platform. Suggestions included the addition of user logs, an audit process by which to judge project quality, progress reports for dissemination to other areas of the organisation and “time stamped” projects to enable projects to be reviewed and updated months or years in the future.

Benefit 3: Builds Evidence. In this trial, organisations did not upload a large quantity of evidence to the ARK Platform, but all three were highly interested in the potential for evidence building. The nature of the CUBE analysis makes it invaluable for producing evidence, for example, by allowing users to translate safety managers’ implicit knowledge and experience into explicit knowledge on the platform by creating a repository of knowledge and information for members of the team or organisation. Participants were interested in developing a repository of past projects as this leads to an improved understanding of issues over both time and place and, thus, enhances the spread and sustainability of change implementation.

Benefit 4: Engages Stakeholders. Partially as a result of the benefits outlined above, participants noted that the ARK Platform has potential to effectively engage a wide range of stakeholders within the organisation. The depth of analysis, inclusion of supporting evidence and focus on social aspects of change were emphasised as key elements of stakeholder engagement. Users hoped to build upon this potential by adding pathways for communicating the information on the platform to the wider organisation. It will be important to find ways of communicating the information within the ARK Platform to those who are unfamiliar with its methods and terminology.

Benefit 5: Shares Knowledge. This trial demonstrated a high level of potential for constructive collaboration across the CoP. The focus group itself led to the sharing of knowledge between organisations. There were many commonalities in terms of experiences using the platform, observations of current impacts, benefits and drawbacks, and visions for future impacts and benefits. Further development of the CoP will allow for benchmarking and standardisation within and between organisations. Participants placed a high value on the ability to share not just data, but also knowledge, between organisations. They explained that this fulfils a need for shared knowledge on the application of standards and practices within healthcare, which is especially relevant for IPC.

Drawback 1: Unclear Workflow. The analysis components of the platform were appreciated for the richness and detail offered, but users noted that some of the sub-questions were abstract and difficult to relate to their own work context. It was hard for users to understand what the sub-questions were asking and which information went where, making the analysis take longer than anticipated and the platform feel overwhelming. A common insight from the three organisations was the need for a road map to assist them in navigating around the platform as they worked on a project. While a technical solution to the signposting issue will be implemented in future versions of the platform, this finding indicates that the training of platform users also should be enhanced to provide a greater understanding of the sequence of progression through a project.

Drawback 2: Lacks Outputs. In the current version of the ARK Platform, no clear outputs are produced. This was identified as a significant drawback for participants throughout the trial in addition to during the focus group. Users suggested a reporting structure to link projects to the rest of the organisation. In particular, organisations would benefit from handover reports between each project stage, a final report at the conclusion of a project and a report on the evidence compiled throughout the project. Users felt that this could be supplemented by an auditing process to measure a project’s progress, with associated progress reports for dissemination to the organisation. In this trial, much work was conducted in terms of designing strategies for transforming the information contained in the platform into action; participants were pleased with the reporting feature currently under development for the next version of the platform and enthusiastic about contributing to the design of other output features.

### 3.3. Evaluation of Need Fulfilment

Table 3 provides an overview of the needs identified in the Introduction, together with the platform feature that most closely addresses that need (Column 2, Verification) and whether or not that feature had been used in the trial (Column 3, Validation). Broadly, the features of the platform that support the cycle of project management were implemented, but the projects themselves were only implemented as far as the problem and solution phases. The features of the platform that support a strategic view through the synthesis of multiple projects have not yet been developed. This sequence of development largely dictates the extent to which the needs have been verified and validated.

Some of the system needs were verified in a platform feature and at least partly validated in the trial: the platform was used to link to a wide range of data (though there has not yet been a great deal of analysis of that data) (1); the CUBE framework was deployed in analysing IPC issues in a way that can address state-related non-linear relationships (2); the CUBE drew in and represented different experiences and points of view—the human role (3). A comprehensive training programme in advanced risk management was developed and an initial module was delivered (8); the trial one projects were initiated in a way that was well embedded in each organisation’s priorities and systems (9); and a framework for data governance was implemented (14).

Those needs addressed by features that were developed, but have not yet been fully implemented in the trial, include the following: The management of risk in change seeks to analyse and facilitate preparedness and readiness for change, recognising that purposeful change involves active self-directed participation as well as leadership (4). The full sequence of project phases from plan to implement to verify gives a project-level basis for action, but this also needs to be scaled up to strategic level (5). There is a strong sequential link from safety to improvement, but the improvement planning and implementation process has not yet begun, and this was not linked to Lean initiatives (13). The development of common organisational capabilities for improvement can be facilitated by a combination of the platform and training (12). A system for data federation was established and tested against ISO criteria (15). Data federation is the first step in enabling the production of sharable knowledge between organisations (11).

The fulfilment of some of the needs depends upon the analysis of multiple cases, the capacity for which is under development. These include the capability to identify emergent characteristics from multiple projects (6), and strategic coherence across many initiatives (7). While the collection of evidence from each case study has begun, systematic evidence collection will be enabled by linking multiple projects in a common knowledge base through machine inference and suggestion capabilities, which are yet to be developed (10).

## 4. Discussion

Given the stage of development of the ARK platform, the results of Trial 1 are encouraging. During and following training, CoP users actively engaged with the platform, and found that it encouraged collaborative analysis addressing a wide range of factors that would not normally be in focus. There are six key issues for discussion, which are explored below.

### 4.1. Platform Use Challenges

One feature reported in multiple trial teams was the time-intensive nature of performing a full STSA of risk and an associated corrective project using the CUBE methodology despite the assistance of the ARK Platform. This was, in part, due to unfamiliarity with the user interface and the CUBE methodology, but it also reflected the deeper and richer analysis of risk that CUBE invited. For example, participants reported that using the CUBE brought in considerations of culture, communications and sense-making that were not often considered before within the organisations. The process also encouraged wider participation in cross-functional discussion of tacit knowledge associated with specific roles within the organisations that participants reported had never been collected into one place before. It is also true to say that we are at the start of the organisational journey with the platform/Mindful Governance process and so as more evidence and project case studies are added, the platform can play a greater role in suggesting similar or relevant material and hence auto-completing or suggesting text or evidence for analysts.

Nonetheless, it highlights the bootstrap problems in any such system and the current lack of output, analytics or reporting on risk or projects for users. In the longer term, automated data integration and knowledge extraction will play a larger role in ARK as ingesting qualitative evidence and turning it into a machine-readable evidence base will allow for more powerful insights. Even with the current evidence ingestion and automated metadata creation features, there was a slow response of the clinical staff to engage with these features (influenced by the delay in DPA and NDA processes). More training is certainly needed, and this need will increase as the automation level increases. Prioritising the most useful reports for implementation was a key goal of Trial 1 (see below for a discussion on reports). For many potential features of the platform, this principle is true: software development is expensive, and our resources are limited, meaning we need to identify the priority areas where we can increase automation, data integration and ease of use (usability) for data search, analytics and suggestions in order to maximise impact. The ongoing trials give us a mechanism to create such a prioritised list. It is important to note that the ARK Platform and CUBE STSA are still human-centric approaches that rely on human expert judgement and governance rather than a fully automated system. A key benefit of this is the ability to be operational without every possible feature being present.

### 4.2. Further Development and Implementation

This paper reports on a particular stage in the development and implementation of the ARK Platform, Trials 0 and 1 of the ARK-Virus project. A large part of the purpose of this phase was to set an agenda for Trial 2, both in terms of both technology development and operational aspects of risk management. In relation to the technology, the evaluation highlighted the type of reports that the system needs to support. The reports have a variety of functions: they help manage the process for a user, recording progress and completeness of use; they manage the relationship between users, for example, in key handovers from those responsible for designing an effective solution (to a problem of importance) to those responsible for a viable plan and for managing implementation, and again to a joint verification of the outcomes of the solution as implemented; the reports also provide an accountable progress and outcome of both an individual project as well as a set or series of projects. The next trial will link draft reports to succeeding stages of implementation.

The next trial will move some existing projects to the next stage of implementation and initiate some new projects of organisational importance. There is a practical organisational challenge in creating a “joined up” organisation in which there is a routine link between the problem and solution defined under the leadership of the risk and safety experts and plan and implementation led by operational management, for example. A new form of risk will be introduced—the RiC process itself. The needs of an extended set of users will be explored in trial two and feed into further development and implementation in trial three.

### 4.3. Data and Legal Aspects

Part of the federated data governance challenge in the trial so far has involved moving the legal agreements between partners to the stage where the risks of personal data processing and confidentiality are adequately understood, using standard contracts, ISO 27000 analysis and data protection impact assessments (DPIAs), to enable legal agreements to be signed to underpin the federation. The ARK Platform’s web-based system architecture has played a role here. As a cloud-based solution, it requires at least some data storage and processing to take place off-site for the CoP members, for example, platform user account names and identifiers. Hence, even with a privacy-aware technological solution, there are legal and compliance hurdles to be overcome and perhaps providing more support for this deployment phase would be beneficial, e.g., generating DPIAs based on project templates and platform capabilities. Of course, health data are one of the most sensitive and legally protected types of data, making the ARK Platform complicated to deploy, but it also made the trial an excellent source of requirements for further development of solutions that will be effective in the clinical setting.

The deployment process also highlighted the importance of providing a sophisticated and flexible security architecture for the platform in order to be able to adapt to individual organisational policies and the needs or norms of data sharing in the community of practice [66]. This is taking place in the context of major upheavals in data sharing both at the technical level where looser dataspace models are increasing in importance [84] and automated privacy support is increasingly part of the agenda for a new generation of data platform architectures such as Gaia X [85] and wider calls for standards to support automation of GDPR compliance [68]. Nonetheless, now that the legal basis is negotiated, the path is clear for further innovation to build upon this agreement and the privacy-aware features present in the platform to enable new features such as confidentiality and privacy-respecting evidence distillation and sharing for community of practice members to gain new insights into risk management and evidence-based best practice.

### 4.4. The CoP

The groundwork for the development of sharable information across a prototypical CoP was laid in three ways: the ARK-virus project itself was a multi-organisational collaborative effort; the participants from the healthcare organisations expressed a strong interest and motivation to share information to develop a common capability to develop and implement effective risk management measures in relation to IPC; and the information architecture was developed to fully protect personal and confidential data. The next stage is to configure sharable outcomes from each project, from which a synthesis of best practice can be constructed. This, in turn, will form the basis of a discussion with regulatory and commissioning agencies about the wider policy implications of this development.

### 4.5. Advanced Risk Knowledge—Competence and Training

The development and implementation of Trial 1 was greatly facilitated by several factors. The principal collaborator in each healthcare organisation had completed a masters in Managing Risk and System Change and the implementation teams contained extensive education, experience and knowledge of social and organisational factors in their particular domain and broader. The advanced risk management training had a particular focus on linking training to implementation and using the ARK Platform as a vehicle to develop the projects and extended participation to the wider implementation teams at the hospital and the fire service. Thus, each organisation had a high level of competence to address the complexities of using the CUBE and ARK Platform to address the risk management of IPC in their organisation. However, other staff without this background were also involved. A further challenge will come as projects go into a wider implementation phase, which will inevitably draw in more people, particularly on the operational side, where the demands of implementation must be balanced against the continuing demands of normal operations. This will draw in further training and facilitation support. The training developed will become available as a regular on-line course contributing to Continuous Professional Development credits, and will contribute to an existing micro-credentialing system leading to Certificate, Diploma and Master’s qualifications.

### 4.6. What Has the ARK Platform Achieved So Far?

The CUBE originated as a theory-heavy, labour-intensive, manually executed and document-centric process for STSA and change. Its completion depended on highly trained risk/safety/organisational change practitioners. It provided insights plus analysis, evidence and knowledge (organisational, domain-specific and discipline-specific), but this was hard to re-use. The ARK platform transformed this in the following ways:A document-centric process is transformed into fine-grained machine-readable formal knowledge models captured as information technology ontologies and taxonomies (knowledge graphs). A common, formal, logical, interoperable framework for risk and safety governance and change project management was defined. This enables cross-project and cross-organisation integration of common project or risk features, evidence, best practice and outcomes.Structured knowledge is captured that enables, for example: the linking of evidence data/datasets to units of analysis; the combining of quantitative and qualitative data in a single model; automated gap analysis and gap filling suggestions; automated tracking and reporting on progress of analysis; quality validation of CUBE data; and the generation of project and risk level reports. It is easy to publish, interlink and share the knowledge as Linked Data on the web. This will lead to richer evidence bases and greater shared experience. Flexible user interfaces enable analysts to quickly view the risks, analysis and evidence from different perspectives (project, risk reporting, CUBE analysis and project stage) in order to more easily synthesise and develop domain knowledge and derive insights into risks, projects and organisational change.In turn, less trained safety experts can apply the CUBE methodology and untrained operational staff and organisational stakeholders can benefit from risk reports and incorporate risk governance into their operational or management activities. CUBE concepts and definitions will be refined through formal modelling. Abstract CUBE concepts will be linked and mapped to concrete risk management (e.g., ISO 31000) processes, artifacts and tools.

## 5. Conclusions

The fundamental argument of this paper begins with a deep-seated systemic problem manifest in terms of persistent stability in levels of quality and safety in the face of accumulating evidence of the need to change. Change initiatives are prone to failure and difficult to sustain and generalize. Where success is reported, it involves long timescales and large organisational effort. An effective response to this challenge must be proportionate to the scale and dimensions of the problem: the analysis of the problem must make possible the formulation of solutions commensurate with the problem’s importance; the solutions derived need to be effective; and the pathways to implementation need to be viable. The ARK platform, supported by training, and as deployed in a set of implementation trials, was put forward as a key mechanism that could engineer leverage over this systemic stasis. The ARK platform is a prototype, still being developed and undergoing its first full-scale implementation trials. These considerations were addressed through three broad research questions.

*In what way and to what extent does the ARK platform meet a set of high-level operational needs for a system dedicated to managing the risks of system change in complex STSs?* The underlying dimensions of the problem space were defined in terms of a diverse set of needs to be addressed, under the following headings: STS complexity and whole system change, evidence concerning healthcare quality improvement, risk and safety governance and governance of data and information. The ARK Platform was assessed against those needs in terms of the features that have been implemented so far in the platform and those that have been further validated in the trial. Broadly, the features of the platform that support the cycle of project management have been implemented, but the projects themselves have only been implemented as far as the problem and solution phases. The features of the platform which support a strategic view through the synthesis of multiple projects have not yet been developed. It is anticipated that the platform will be able to address each of the identified needs over the next stage of development.

*Is the proposed solution effective in practice? Does the framework built into the ARK Platform enable the adequate analysis of a complex operational problem and solution space?* Two of the three organisations were able to effectively initiate a project using the ARK platform. Their experience was that the platform, supported by the training, expands the depth and breadth of risk management, supporting the transparency of the process and engaging stakeholders, with the potential to build evidence and share knowledge. This has initiated a process of collecting evidence in machine-readable form which is the first step in building linked data in an evidence base that can support synthesis across multiple projects and produce sharable knowledge.

*Is there a viable pathway to implementation? In this case, the focus is on the architecture and interface design: How well does the ARK Platform support the user in carrying out these tasks?* There are several usability issues, including complex questions, unclear workflow and a lack of outputs (reports) from the platform. Performing a full STSA of risk and initiating a corrective project is time intensive due, in part, to unfamiliarity with the user interface and the CUBE methodology, but also reflecting the deeper and richer analysis of risk that the CUBE invited. Users consider the ARK Platform to be useful and to have great potential for healthcare risk governance. The feedback will be used to further develop the ARK Platform in preparation for Trial 2 of the ARK Virus Project.

The pathway to implementation is in its early stages. These initial results are encouraging in terms of the active engagement of participants with the ARK Platform addressing the complexity of the operational system, and informative in providing guidance for the next stages of system development. These next stages set out a challenging research and development agenda to consolidate the platform’s capabilities in coping with the complexities of data and analysis in an integrated way, fully supporting adaptation processes (implementation and change) and developing more powerful system governance capabilities. Moving from analysing problems and devising solutions to implementing those solutions and verifying the outcome will test both platform development and the participating organisations: risks change over time and, therefore, priorities for implementation may change; wider participation of operational staff in the implementation phase will challenge the organisations and test the efficacy of the risk in change concept; and further training will be needed. Critical research questions concern the right trade-off between the complexity of the analytic framework, the competence levels of different users and the requirement to progress interventions in a timely and accountable manner.

The next trials will improve the outputs of the ARK system, improve its capability to manage multiple projects and integrate diverse sorts of evidence and actively support phases of implementation. Populating the platform will enable the development of the knowledge-based capabilities that will enable better support for the user in accomplishing individual complex tasks at the same time as building and synthesising a growing evidence base that supports risk governance at system level. This will increase the motivation to share evidence, to build better practice in common and to share with the wider healthcare community. This will require alignment of the platform configuration, data governance, and the organisation of the CoP and links with the wider healthcare community including regulatory and commissioning stakeholders. As it develops, this will generate a need for new sustainable knowledge-based services to support a growing CoP. Critical research questions here include the configuration of the ontologies that manage the data and support the user to construct the knowledge about dynamic organisational processes that have consequences.

The core proposition is that the opportunity offered by technologically sophisticated knowledge engineering systems (such as the ARK Platform) can fill a critical gap in the innovation cycle. Developing the knowledge base as a practical tool makes possible the building of both competence and know-how (on the one hand) and the governance of evidence-based best practice (on the other), not just at the intervention level but also at the system level. Such knowledge provides the basis for improved leadership, planning, resource allocation and development. Additionally, if shared, it enables people to make sense of their own situation and understand their strengths and limitations, and can form the basis of effective participation in change. Ultimately, such knowledge can become embedded in the culture of the organisation as it becomes part of its everyday life and routines. Such a knowledge system, therefore, has the potential to help reduce the uncertainty and high failure rate of interventions to improve the quality of the health system and to reduce the time it takes for successful systems of organisational transformation to become mature. This creates a powerful research agenda as well as a practical programme of action. The goal is, therefore, to continue to build an infrastructure to engineer knowledge-based solutions to hitherto apparently intractable problems and to support their implementation and governance in a way that is both sustainable and generalisable across the system.

## Figures and Tables

**Figure 1 ijerph-18-12572-f001:**
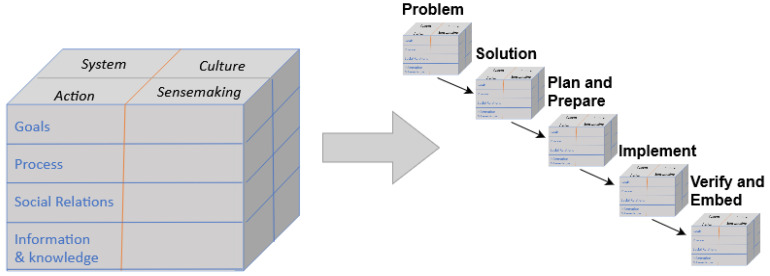
Dimensions of the CUBE.

**Figure 2 ijerph-18-12572-f002:**
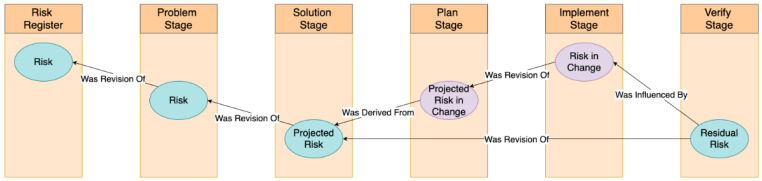
Transformation of Risk through project stages in the ARK Platform.

**Figure 3 ijerph-18-12572-f003:**
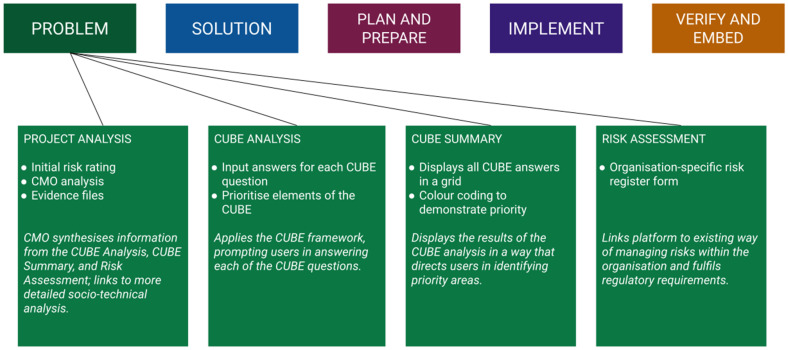
Risk and Project Analysis views in the ARK Platform.

**Figure 4 ijerph-18-12572-f004:**
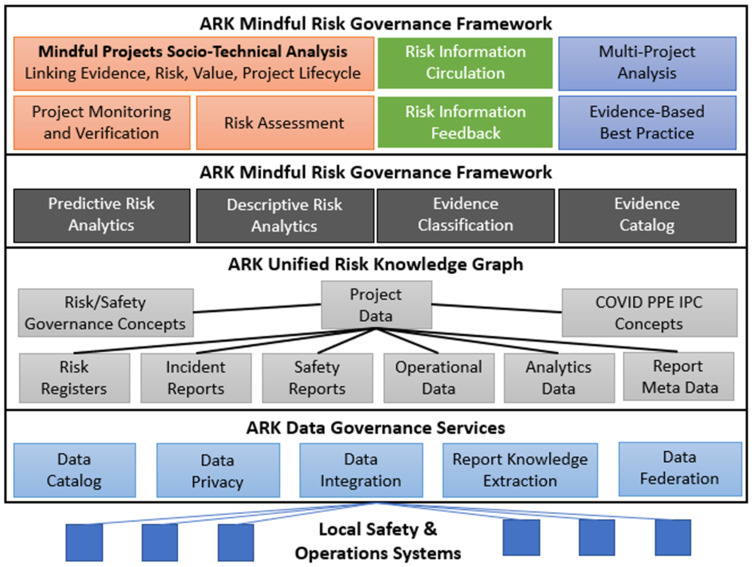
ARK Platform Mindful Risk Governance Framework.

**Figure 5 ijerph-18-12572-f005:**
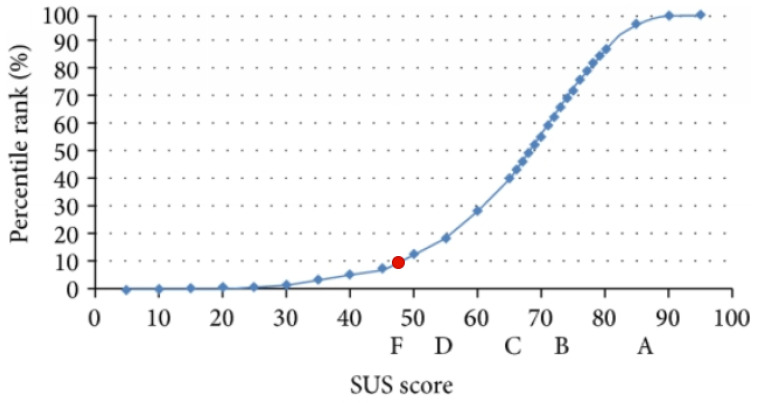
SUS Score percentile ranking. The red dot represents the average score given to the ARK Platform by participants.

**Table 1 ijerph-18-12572-t001:** ARK-Virus Project phases and research activities executed/planned.

Platform Development Trial	Research Activities	Methodology
0	0.1 Coalition building 0.2 Requirements gathering	Regular discussion between organisational psychology team, software engineering team and collaborating healthcare organisations
1	1.1 Requirements gathering 1.2 Platform enhancement 1.3 Platform deployment and user evaluation 1.4 Evaluation according to identified needs	Questionnaires, monthly meetings, project initiated on platform within each organisation, qualitative and quantitative user feedback, tracking of platform use metrics, validation according to operational risk management needs
2	2.1 Platform enhancement based on new requirements 2.2. Platform deployment and user evaluation	Continuation of projects initiated in Trial 1 through the plan and prepare phases, user feedback, continued assessment of fulfilment of operational risk management needs even as needs evolve
3	3.1 Platform enhancement based on new requirements 3.2 Platform deployment and user evaluation	Continuation of projects through the implementation and verification phases, user feedback, continued assessment of fulfilment of operational risk management needs even as needs evolve

**Table 2 ijerph-18-12572-t002:** ARK platform metrics.

Question	No. of Participants				
	1 (Not at All Useful)	2	3	4	5 (Very Useful)
How useful was the ARK Platform in your organisation?		1	2	2	1
How useful was the Project Analysis component of the platform?			3	3	
How useful was the CUBE Analysis component of the platform?			1	3	2
How useful was the CUBE Summary component of the platform?				4	2
How useful was the Risk Assessment component of the platform?			3	1	2

**Table 3 ijerph-18-12572-t003:** Elements of a fully systemic approach to healthcare change management. Bolded text indicates the extent to which each element has been addressed in the ARK Platform (either verification or validation).

Needs	ARK Feature—Verification	Implementation—Validation
**Socio-technical System Needs**		
1. Multiple interacting causes and consequences	**Evidence section of the platform enables links to diverse data sources.**	**A wide range of relevant data are being collected. These will be essential in defining the performance aspects of each project as they continue.**
2. Non-linear relationships	**CUBE STA analysis with four dimensions** **of complex problem and solution spaces.**	**Evidence from several dimensions have been gathered in the analysis, capturing non-linear relationships.**
3. The role of people	**CUBE enables different points of view (POVs) to be included, represents experiences from different stakeholders and cultural aspects of the system.**	**Different stakeholders in the project, e.g., quality managers, safety managers and systems improvement managers, gave input from their work perspectives and roles.**
4. Self-organising tendencies of adaptation and change	**Plan and Implementation phases support analysis of factors contributing to risk in the change process.**	People working together on a project, sharing understanding and preparing purposeful implementation. Project implementation phase not yet tested.
5. Adequate basis for action	**Risk and value parameters built into each project stage, problem, solution, plan implement, verify. Strategic priorities not yet supported.**	Risk assessed at problem and solution phase but follow through in implementation not yet conducted.
6. Emergence	**Capability to synthesise multiple projects is under development.**	Common findings despite diverse approaches underline commitment to explore common approach to benefit all organisations.
7. Strategic coherence	Risk and value frameworks implemented at project level, but no system-level risk register yet.	Requirements gathered for strategic synthesis of multiple projects.
**Healthcare quality improvement**		
8. Training and education	**Advanced Risk Management training developed includes introduction to CUBE and projects, management of advanced data analytics, operations management, measuring and monitoring risk and safety, change and improvement, governance and strategy.**	**First module of training delivered. Some in implementation teams have extensive socio-technical knowledge; for others, it is new.**
9. Improvement processes embedded within normal management activity	**Platform incorporates the risk management form used in the Irish health system. Trial 1 enhances normal risk management process for IPC.**	**Each organisation’s project was refined to fit within their existing priorities, systems and processes.**
10. Provides a systemic methodology for collecting evidence	Analysis and evidence are interlinked. This will enable machine inference and suggestion to be developed, linking multiple projects in a common knowledge base.	Trial 1 begins the population of the platform with related projects.
11. Produce shareable knowledge within and between organisations	**Data governance rules established. Data are processed in compliance with General Data Protection Regulation and confidentiality.**	Initial CoP works well. Common ontologies to be further developed to relate knowledge domains to each other—this will enable sharing within GDPR.
**Risk Governance Needs**		
12. Common organisational capabilities for development, improvement and change	**Advanced risk management training and platform use can foster organisational capabilities for development, improvement and change.**	Strong mobilisation of effort for defining problem and solution. Not yet moved into implementation. Training focused on improvement not yet delivered.
13. Integrated approach between safety, risk and Lean	**Links risk management to a powerful implementation and change framework.**	No engagement with Lean improvement initiatives
**Data Governance Needs**		
14. Data governance infrastructure	**Platform creates metadata to operationalise data governance. Embedded data governance service for evidence management. Defined terminologies.**	**Data catalogues to find data, data quality to assure the data, data security model to enforce policy.**
15. Privacy-aware Data Federation (reuse/interoperability across organisations)	**Each organisation has a protected area of the platform. All data classified as Public, Internal, Confidential or Restricted depending on sensitivity, inter-organisation sharing policies. Data protection-oriented data linking across organisations. Risk, change and health domain ontologies for common models.**	Data transformation removes private, confidential content and connects data silos. ARK Platform evaluated against the ISO 27001 Information Security Management Standard (https://www.iso.org/isoiec-27001-information-security.html (accessed on 23 November 2021)) and ISO 27701 Privacy Information Management Standard. (https://www.iso.org/standard/71670.html (accessed on 23 November 2021)) and received high compliance scores of 85% and 91%, respectively.

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
