# Peer review of "Evaluation of an Access-Risk-Knowledge (ARK) Platform for Governance of Risk and Change in Complex Socio-Technical Systems"

_ijerph, 2021, doi:10.3390/ijerph182312572_

Round 1

Reviewer 1 Report

The authors present  an approach to the management of risk, safety, improvement and change that incorporates an existing framework - the CUBE, and the ARK (Access Risk Knowledge) Platform. The implementation and application of a platform (ARK) aims to support the fully cycle of implementation of improvement projects. A survey was conducted to evaluate the usability of the platform using fifteen needs as major aspects related to complex system transformation. Furthermore, the implementation is in the trial phase while the platform is still under development.

The manuscript is very interesting, however due to the huge amount of diverse information it is very difficult to clearly understand the message that the authors want to deliver.

In my opinion when describing point 1.5. Elements of a Fully Systemic Approach to Change: Needs Gathering, (a, b, c, and d ) a table should be used to describe the sub points. This would facilitate the reading and understanding.

I think that the subject addressed in the manuscript is of high value for healthcare organizations which could be applied also in other industries. However, in my opinion it should be more focused in the key message that the authors want to deliver. I would suggest the authors to be clearer with the purpose of the manuscript and to framework the major steps that represent the major steps conducted. For example, create a Table with the major steps regarding the development, implementation, and evaluation of the proposed model.

Because the work is quite extensive (for example the introduction chapter is to massive), I think it would be beneficial to introduce a sub-chapter after in the introduction chapter to illustrate how the work is structured. For example, 1.2 Structure of this Work and Relevance and Novelty of the Research.

The information in the conclusions chapter  could be also  divided into four major blocks: For example:

  1. Conclusions,

5.1 Academic Implications,

5.2  Managerial Implications and

5.3 Suggestions for Future Research

Finally, some minor aspects that regard the structure-standard of the manuscript should be addressed.

  • The referencing /citation style across the document is not according to the MDPI standards. Please correct. Also, the references in the References

  • “chapter” are not according to the MDPI standard. Please correct.

  • In the introduction chapter point 1.6 is not mentioned, jumping from 15 to 17.

  • Figure 5 should be enlarged to better see the concepts and the relationships of the ARK (Access Risk Knowledge) framework.

  • Tables are not formatted according to the MDPI standards. Please correct.

Thank you very much.

Reviewer 2 Report

Comments to the Author

This paper proposes a novel possible solution for accessing the knowledge platform with the perspective of risks and system changes.

The introduction part is very general. Also, no specific argumentation is given why this approach is suitable for this research, or how it relates to others.

  • There is a gap between the research background and the specific issue discussed in this study.
  • The concepts and root causes are not supported well in relevance to the specific issue of this study. Instead of just describing the contents, it is necessary to point out the logic and interconnection among the concepts used.

In the methodology part, the proposed approach does not emphasize the importance of the research topic and solution.

Small remark on the finding section, it contains a collection of many results, but it is important to select only the relevant results related to the theme of this paper in sequential order.

  • The tables used in the finding section have poor descriptions, it must explain the importance and necessity of the work and collected data to support the work.

In conclusion, this paper could be improved by restructuring the story of this paper that includes only the relevant information.

  • Research questions are missing in this paper that could give an important direction to the solution by answering the questions as results.

Reviewer 3 Report

Abstract: Access Risk Knowledge needs to be fully used in the abstract, not the (ARK) abbreviation.

Introduction: lines 60-65, 69-71 = quotations, not required, remove or rephrase.

line 93 - 'this paper argues', please try to remove anthropomorphism.

Figure 1. text is not clear/use another font, action of 'intends' is not clear in meaning

Figure 3, 4, 5. font size is too small

Lines 964, 965 - no need for red font

Table 3, red font is visually not appealing

Discussion, no need to bullet the key issues if used as subheadings.

Round 2

Reviewer 2 Report

Thank you for the detailed revision.

The paper is reasonably written in terms of language usage and is structurally well presented. However, there are still some typos and grammatical errors. Please do thorough grammatical checking of the manuscript to improve the readability of this study.

Author Response

On behalf of the research team, thank you very much for your feedback. We have completed a full review of the manuscript for typos, grammar, etc.